# Sensing, Signaling, and Secretion: A Review and Analysis of Systems for Regulating Host Interaction in *Wolbachia*

**DOI:** 10.3390/genes11070813

**Published:** 2020-07-16

**Authors:** Amelia R. I. Lindsey

**Affiliations:** Department of Entomology, University of Minnesota, St. Paul, MN 55108, USA; alindsey@umn.edu

**Keywords:** symbiosis, endosymbiont, two-component system, type I secretion system, type IV secretion system, gene regulation, transcriptional regulation, environmental response

## Abstract

*Wolbachia* (Anaplasmataceae) is an endosymbiont of arthropods and nematodes that resides within host cells and is well known for manipulating host biology to facilitate transmission via the female germline. The effects *Wolbachia* has on host physiology, combined with reproductive manipulations, make this bacterium a promising candidate for use in biological- and vector-control. While it is becoming increasingly clear that *Wolbachia*’s effects on host biology are numerous and vary according to the host and the environment, we know very little about the molecular mechanisms behind *Wolbachia*’s interactions with its host. Here, I analyze 29 *Wolbachia* genomes for the presence of systems that are likely central to the ability of *Wolbachia* to respond to and interface with its host, including proteins for sensing, signaling, gene regulation, and secretion. Second, I review conditions under which *Wolbachia* alters gene expression in response to changes in its environment and discuss other instances where we might hypothesize *Wolbachia* to regulate gene expression. Findings will direct mechanistic investigations into gene regulation and host-interaction that will deepen our understanding of intracellular infections and enhance applied management efforts that leverage *Wolbachia*.

## 1. Introduction to *Wolbachia*

*Wolbachia pipientis* is an alpha-proteobacterium within the Anaplasmataceae, a family of anciently intracellular bacteria belonging to the Rickettsiales [1]. The Anaplasmataceae family includes well known intracellular pathogens such as *Rickettsia*, *Anaplasma, Erlichia,* and *Orientia*, as well as *Midichloria*, which reside within the mitochondria of certain ticks [2,3]. *Wolbachia* are somewhat unique within the Anaplasmataceae: they are not transmitted to vertebrate hosts via an arthropod vector like many of the pathogenic species. Instead, *Wolbachia* colonize many species of arthropods and nematodes and are transmitted from mother to offspring via the maternal germline [4,5]. Additionally, *Wolbachia’s* interactions with hosts are highly variable, and include pathogenic and mutualistic effects, even within the lifespan of a single *Wolbachia*-host association [6,7,8].

*Wolbachia* were initially categorized as “reproductive parasites” due to their colonization of insect germlines, inclusion bodies reminiscent of other intracellular infections (i.e., viruses), relationship to the pathogenic Rickettsia, and the reproductive phenotypes induced in hosts, including skewing sex ratios and creating sperm-egg incompatibilities [9,10,11,12]. However, it is becoming clear that manipulating the reproductive biology of hosts is not a feature of all *Wolbachia* strains, nor is it essential for the spread of *Wolbachia* through a population of hosts [13,14]. Despite the absence of reproductive manipulations in many *Wolbachia*-host associations, *Wolbachia* are effective at spreading throughout a population of hosts [14,15]. Indeed, many populations of arthropods and nematodes are fixed for *Wolbachia* infection, and an estimated 40% of arthropod species have *Wolbachia* infections at some level [16,17]. Proposed mechanisms for the advantage *Wolbachia* provides that would result in spread through a population include metabolic provisioning [18], higher fecundity relative to *Wolbachia*-free hosts [8], and protection against viruses [19]. Additionally, while *Wolbachia* are primarily transmitted from mother to offspring, they are generally capable of host-switching and establishing within a new, phylogenetically distant host species, as is evident by highly incongruent host and *Wolbachia* phylogenies [11,20].

Again, in contrast to the pathogenic members of the Anaplasmataceae, we know comparatively little about the mechanisms *Wolbachia* uses to interact with hosts and manipulate their biology. *Wolbachia* studies across the past few decades have been overwhelmingly focused on evolutionary biology and genomics [21,22,23,24], anti-viral protection [19,25,26,27,28,29,30], or molecular biology of host factors [31,32,33], rather than bacteriology of *Wolbachia*. This is in part due to the combination of research groups that have studied the *Wolbachia*-host symbiosis over the years, and the availability of genetic tools for host insects (e.g., *Drosophila*) and not for *Wolbachia.* However, within the last couple of years, several key studies have been published that provide some insight into the mechanisms *Wolbachia* uses to interface with host biology. Namely, the first link between *Wolbachia* genotype and phenotype was identified [34], the *Wolbachia* loci that induce and rescue sperm-egg incompatibilities (known as “cytoplasmic incompatibility” or CI) have been identified and enzymatically characterized [35,36,37], a suite of putatively secreted effector proteins have been identified and made available as vector constructs [38,39,40], and significant attention has been given to the potential functions encoded within *Wolbachia*’s prophage, “prophage WO”, beyond CI [41,42]. While these studies are particularly exciting and will be beneficial for directing applied efforts that leverage *Wolbachia* for insect management, a vast majority of *Wolbachia* genes are uncharacterized. Additionally, very little is known about what pathways and loci are critical for colonizing hosts and manipulating their biology. Furthermore, how *Wolbachia* regulates these interactions across host life spans or upon encountering novel hosts is almost completely unknown.

While obligately intracellular bacteria in general have more stable environments (i.e., the host cell) than free-living bacteria, even bacteria that are the most reduced in complexity and are obligate for their host have maintained some level of regulatory capacity [43]. For example, in *Buchnera*, the highly reduced nutritional symbiont of aphids, there is differential protein expression that appears to be regulated by small RNAs [44]. While *Wolbachia* have relatively reduced genomes (typically between 0.9 and 1.5 Mbp [45,46]) and they are obligately intracellular, *Wolbachia* are not obligate for the host in most circumstances, and they are not as strictly maternally transmitted as some obligate symbionts such as *Buchnera.* Thus, the ability to respond to their host environment is likely to be even more critical for establishment and transmission.

## 2. How do Other Bacteria Interface with Their Environment?

In facultatively intracellular bacteria that transition between environmental and host-associated states (e.g., pathogens such as *Shigella* [47], *Brucella* [48], *Francisella* [49], *Mycobacterium* [50], *Salmonella* [51], and *Legionella* [52]) and symbionts that regularly alternate between intracellular and extracellular states within a host (e.g., *Spiroplasma* [53]), bacteria must be able to sense whether or not they are within a host cell and mount an environment-appropriate response (e.g., virulence, growth and division, changes to metabolism, persistence, etc). Obligately intracellular bacteria (including *Wolbachia* and other Rickettsiales, *Coxiella* [54], and *Chlamydia* [55]) have a much more stable environment than free-living or facultatively intracellular bacteria, but must still be able to respond to changes in their surroundings. Regulation in response to host signals is important for (1) the bacterium to modify its own physiology, and/or (2) for the bacterium to exert reciprocal effects back on host physiology. Ultimately, this might allow a bacterium to avoid immune responses, regulate their metabolism and virulence, and alter host physiology in other ways that enhances their own fitness. To accomplish this, bacteria must be able to (1) detect important intra- and extracellular signals (e.g., changes in the bacterium’s own nucleotide pool, or the presence of an antimicrobial peptide), (2) relay that signal appropriately, (3) modulate gene and protein expression, and (4) properly traffic and/or export gene products and metabolites.

In bacteria, including those that live intracellularly, signals might include mechanical contact, pH, temperature, osmotic pressure, or the presence of metabolites, toxins, and proteins, all of which can be sensed by a variety of receptors [50,56,57,58]. These receptors either work alone (as a one-component signaling system) or they may be members of two- or three- component signaling systems that involve other proteins to relay the signal and initiate changes in metabolism or on gene expression [59,60]. One-component signaling systems typically involve an intracellular protein that responds to a stimulus and causes a response, with the input and output functionalities encoded in different domains [59]. Two-component systems (TCS) are the canonical pathways that bacteria use to link the environment to gene regulatory responses [61]. TCS employ a transmembrane histidine kinase that autophosphorylates upon receiving a signal, and a cognate response regulator that typically has enzymatic or transcription factor activity [62]. Finally, three component systems work similarly to TCS, except that signal reception is encoded on a separate transmembrane protein which then activates an intracellular histidine kinase [59,60]. Bacteria regularly encode for dozens of these systems and are highly specific for a given input and the downstream response [63]. For example, in *Escherichia coli*, there are dozens of pairs of histidine kinases and cognate response regulators that are critical for sensing and mounting responses to stimuli such as nitrite, phosphate, cell envelope structure, and osmolarity [59]. In host-associated bacteria, TCS are essential for regulating the secretion of toxins and effector proteins, internalization into host cells, trafficking within a host cell, modifying host metabolism, and egress [47,48,64,65,66]. Because TCS sensor histidine kinases are embedded in the inner membrane, factors such as outer membrane permeability affect the ability of TCS to receive and transduce signals [67].

After receiving and transducing a signal, the cell will initiate a response. The response could be at the protein level (e.g., levels of a metabolite are high enough, and negative regulation shuts off the enzyme making said metabolite), or at the level of gene expression. In one-component systems, the response is enacted by the same protein that received the signal [59]. In some two- and three-component systems, the response regulator has transcription factor activity. In some intracellular bacteria, groups of pathogenicity genes are regulated by a single factor (that may or not be a member of a TCS). For example, in *Francisella*, MglA regulates the expression of a genomic island that is required for intracellular growth [49]. In *Legionella*, amino acid starvation seems to induce virulence via an increase in RpoS, the sigma factor that guides RNA polymerase to initiate transcription at specific sites [52].

Downstream effects on virulence and pathogenicity are often mediated through the regulation of secretion systems. Secretion systems are membrane-embedded protein complexes that transport substrates across the membrane into a host, the environment, or into a neighboring bacterium [65,68]. In pathogenic bacteria, secretion systems are key to the success of the infection: the pathogen secretes effector proteins into the host that affect numerous physiological processes, ultimately enhancing the pathogen’s fitness. These secretion systems are tightly regulated by two-component systems and other transcription factors [69,70,71,72,73]. Bacteria regularly encode for multiple secretion systems, with each system specialized to transport substrates. For example, *Coxiella* encodes for a Type I Secretion System (T1SS), a Type II Secretion System (T2SS), and a Type IV Secretion System (T4SS) [54]. *Francisella* encodes for both a T2SS and a T4SS [49]. Many secretion systems are homologous to other structures evolved to span membranes. Type III Secretion Systems (T3SS) were initially discovered in *Yersinia pestis*, are found across many highly pathogenic bacteria, and are related to the basal body of flagella [74]. Several kinds of T4SS are related to bacterial conjugation machinery and some are capable of secreting DNA in addition to effector proteins [75,76,77]. Type VI Secretion Systems (T6SS), discovered in *Vibrio cholerae* and *Pseudomonas aeruginosa*, are highly similar to the tail spike of T4 bacteriophage [78,79,80].

These secretion systems use many different mechanisms to move substrates across membranes. The T1SS is a relatively simple system, encoded by only three proteins: an ABC transporter, a membrane fusion protein (MFP), and an outer membrane protein (OMP) (Figure 1) [81]. These proteins form a tunnel when the ABC transporter in the inner membrane interacts with the substrate, allowing the substrate to pass across the periplasm and out of the cell [81]. Like T1SS, the T3SS, T4SS, and T6SS also transport substrates from the cytoplasm to the outside of the cell (or into a target cell), and span both membranes and the periplasm. However, these secretion systems are made up of many more protein subunits. T3SS, related to flagella, move proteins through a needle-like “injectisome” after chaperones deliver substrates to a portal on the cytoplasmic side of the inner membrane [82]. T4SS are typically composed of 12 protein subunits present in variable copies that form a pilus, a translocation channel scaffold, and ATP-ases that drive a substrate through the channel (Figure 1) [75]. One of the ATP-ases, known as VirD4 in many T4SS, is the “coupling protein” that is responsible for substrate recruitment [75]. T6SS, which functions as an inverted phage tail, exports substrates via polymerization, which pushes the substrate out of the cell [80]. Finally, the T2SS and Type V Secretion Systems (T5SS) span the outer membrane and move substrates from the periplasmic space to the exterior of the cell. In T2SS this is thought to be mediated by a pseudopilus that functions as a piston, pushing a substrate out of the cell [83]. T5SS are autotransporters: they have a β-barrel domain that embeds in the outer membrane and facilitates movement of the passenger domain(s) across the membrane [84].

In addition to secretion systems that span the outer membrane of the bacteria, there are additional systems that translocate proteins from the cytoplasm into the inner membrane, or through the inner membrane into the periplasm [86]. The general secretion system (Sec) directs unfolded proteins through a channel in the inner membrane, composed of Sec proteins Y, E, ang G. Proteins are directed to the SecYEG complex via either (1) a SecA ATPase motor protein that leverages SecB as a chaperone, or, (2) a signal recognition particle ribonucleoprotein complex (SRP) in combination with YidC (an insertase) and a membrane receptor FtsY [86,87]. Additional proteins, SecD, SecF, and YajC, enhance the efficiency of translocation through SecYEG. Finally, signal peptidases embedded in the inner membrane will cleave signal domains off the polypeptide, releasing the mature protein [88]. A separate system, twin-arginine translocation or “Tat” moves folded proteins across the inner membrane [86]. The number of Tat proteins that a given bacterium encodes for and requires for functional Tat secretion is variable [89]. The most minimal Tat systems contain two proteins: TatA and TatC [90]. Other systems use a third protein, TatB, and some bacteria encode for a paralog of TatA, named TatE [90]. All Tat components are embedded within the inner membrane. Folded proteins with the Tat signal sequence are bound by TatC (or a TatBC complex) which then recruits TatA protomers to form a translocation site in the membrane [90]. The aforementioned T2SS and T5SS only span the outer membrane of the bacterium, and thus rely on Tat and Sec systems for the initial translocation of effectors from the cytoplasm to the periplasm [83,84,88]. In addition to secretion outside of the bacterial cell, the Sec and Tat systems are important for embedding proteins in membranes. This includes proteins destined to the inner membrane (e.g., histidine kinases of a TCS), and proteins destined for the outer membrane that are localized to the periplasm via Sec/Tat and then located to the outer membrane by other pathways. These outer membrane proteins may directly interact with host factors [88].

Across the Rickettsiales, many of these bacterial systems have been identified and play roles in establishment, pathogenicity, and transmission. Genes encoding TCS have been identified across all genera in the Rickettsiales (*Rickettsia*, *Anaplasma*, *Ehrlichia*, *Orientia*, *Neorickettsia*, *Wolbachia*, *Midichloria*, and the newly described *Aquarickettsia*), and in some cases have been shown to be differentially expressed across development [91,92,93,94,95,96,97,98]. Several transcription factors have been identified in Rickettsiales that are differentially regulated according to host context (e.g., in the vertebrate host vs the arthropod vector) [92], and there are clear transcriptional responses of Rickettsiales species to stress, temperature, and the host [99,100]. Finally, all members of the Rickettsiales encode for a T4SS which was likely acquired by the ancestor of the clade, and facilitated the evolution of intracellularity [101].

As compared to the other Rickettsiales, relatively few studies have looked at *Wolbachia*’s ability to sense and respond to the environment. However, it seems the field of *Wolbachia* research is seeing a shift as more and more studies are focusing on molecular and cell biological aspects of the *Wolbachia*-host relationship, and of *Wolbachia* bacteriology [102]. This is particularly exciting given the rich understanding of the evolution and natural history of *Wolbachia* across arthropods and nematodes, which we can now begin to link to a mechanistic understanding of the host-*Wolbachia* interface. To facilitate a better understanding of the mechanistic basis of host interaction in *Wolbachia*, I characterize the presence of two-component systems, transcriptional regulators, and secretion systems across 29 *Wolbachia* genomes, and discuss what is currently known about how *Wolbachia* interfaces with its host.

## 3. Materials and Methods

Twenty-nine *Wolbachia* strains with complete or near-complete genomes were selected for analyses. These strains belong to Supergroups A, B, C, D, E, F, and L. *Wolbachia* genomes and their annotations (all annotated with NCBI’s PGAP [103]) were downloaded from RefSeq [104] on 02/10/2020 (for accession numbers, see Appendix A). I queried annotations for proteins involved in sensing and signaling, transcriptional regulation, and secretion using a combination of annotation mining, orthology, BLAST, and manual curation. First, annotations were searched for terms related to the processes (for code, see Appendix A). Then, previously published clusters of orthologous proteins across *Wolbachia* [38,105] were used to annotate proteins and ensure that proteins were not missed due to any differences in annotation. Manual curation further aided in grouping proteins based on domain structure available from NCBI. Previously published data on genes of interest (TCS [95], T4SS [106]) were cross referenced to ensure I recovered expected proteins. Pseudogenization and partial annotations are derived from the PGAP/RefSeq annotations.

Proteins annotated as T6SS components were manually inspected and found to be mis-annotated. They were all located within predicted T4SS operons, had more recent NCBI annotations that re-named these proteins as T4SS components, were homologous to T4SS proteins, and/or had conserved domain structure of either the VirB3 or VirB4 proteins of the T4SS. Given the presence of T1SS proteins HlyD and a T1SS ATPase, I specifically searched for the presence of an outer membrane protein that would complete a T1SS and indicate the potential for it to be functional. A protein annotated as TolC was identified in single copy across the majority of *Wolbachia* strains. Strains missing a protein annotated as TolC had a single putative TolC (labeled as “membrane protein”) that was homologous to the other TolC *Wolbachia* proteins, as inferred by previously published ortholog clustering [38,105].

Plotting was performed in R [107] version 3.5.0 (23-04-2018) “Joy in Playing” with the heatmap.2() function from the gplots package [108]. Figure polishing and schematics were created in Inkscape (https://inkscape.org/).

## 4. Sensing and Signaling

I identified two sets of TCS across the 29 *Wolbachia* genomes: PleC/PleD and CckA/CtrA (Figure 1). These are relatively highly conserved in their presence across *Wolbachia*, with just a handful of losses or pseudogenizations. These findings are in agreement with previously published data on the evolution of TCS across 12 strains of *Wolbachia* [95]. Here, with the addition of many more genomes, a few patterns begin to emerge.

Most losses are in the PleC/PleD TCS, where PleC is the sensor histidine kinase and PleD is the response regulator. PleC typically has enzymatic activity that regulates levels of cyclic diguanylate (c-di-GMP) in the cell, a ubiquitous bacterial second messenger that is involved in the regulation of many complex processes [109,110]. While the signal that PleC detects is unknown, the PleC/PleD system is present in many bacteria and is a key component of cycle regulation [111,112]. Here, I identify six losses or pseudogenizations in the PleC/PleD system, all of which are restricted to the monophyletic clade composed of the C and F Supergroups (represented by the strains *w*Oo and *w*Ov from Supergroup C, and *w*Cle from Supergroup F). Both *pleC* and *pleD* are pseudogenized in *w*Oo and *w*Cle, and in *w*Ov, *pleD* is annotated as pseudogenized while no *pleC* was recovered (Appendix A).

The second TCS present in *Wolbachia* is the CckA/CtrA system, where CckA is the sensor histidine kinase, and CtrA is the response regulator. Again, the signal for activation of CckA is unknown, but the downstream response regulator CtrA is a well described master regulator with transcriptional factor activity that is present across alpha-proteobacteria [112,113,114,115]. I identified two instances of pseudogenization of *cckA*: in *w*Mel and in *w*AlbB (Appendix A). The *w*Nfe strain had two copies of *cckA*, one predicted to be complete and functional, and the other partially present which may indicate an assembly or annotation issue (Appendix A).

The PleC/PleD and CckA/CtrA TCS are present across other Rickettisales, many of which encode for a third TCS not found in *Wolbachia*: NtrY/NtrX, which is involved in the regulation of nitrogen fixation [116]. In *Anaplasma* and *Erlichia*, the PleC/PleD and CckA/CtrA systems are differentially expressed, are enzymatically active (histidine kinases are capable of phosphorylation and PleD can make c-di-GMP), and they have been implicated in regulating intracellular infection [116,117,118]. The role of these TCS in *Wolbachia* biology remains to be determined, but their strong conservation across *Wolbachia* strains indicates these TCS are likely critical for *Wolbachia*’s fitness.

## 5. Transcriptional Regulation

Transcription in bacteria is a highly regulated process and includes three major steps: (1) initiation, (2) elongation, and (3) termination. Initiation involves RNA polymerase (bacteria have one RNA polymerase, whereas eukaryotes have three), sigma factors (“σ factor” or “specificity factor”) which are unique to bacteria, transcription factors (TF), specific DNA sequences where proteins bind, and signals that interact with the proteins (e.g., small molecules or other activating or repressing proteins). Transcriptional regulation in bacteria is more nuanced than described here, but a brief introduction aids to contextualize the genes that *Wolbachia* encode for that regulate these processes.

In bacteria, RNA polymerase requires a σ factor for proper and specific binding to a promotor. Bacteria typically encode for a “housekeeping” or “primary” σ factor (in *E. coli* this is σ^70^—RpoD), as well as additional “alternative” σ factors which are specialized for responses to particular environmental conditions (e.g., heat, stress) [119]. *Wolbachia* all encode for one copy of a RpoD-like protein (in *w*Ppe the copy is pseudogenized, but we cannot rule out assembly or annotation artifacts) (Figure 2). Additionally, all strains have one copy of a σ^32^-like protein, RpoH, which in other bacteria is the heat shock σ factor (in the *w*VitB assembly this gene is only partially present, which may again be an assembly-related problem). Beyond that, some *Wolbachia* strains encode for up to four additional σ factors, all in the size range of 158–181 amino acids long. While the target genes for *Wolbachia*’s σ factors remains to be determined, the presence of multiple σ factors in varying copy number indicates that *Wolbachia* likely controls its transcriptional response to environmental conditions.

TF’s play a role in gene expression via regulating RNA polymerase/σ factor complex binding. TFs typically occlude the promotor region, preventing RNA polymerase and the σ factor from binding, or they bind upstream of the promotor to aid in RNA polymerase/σ factor complex recruitment [120]. TFs typically have two domains: one that binds signals, and one that binds to the DNA at a Transcription Factor Binding Site (TFBS) [120]. TFs may work in concert with each other, some have the ability to bind to multiple sites, and many TFBS can each be bound by multiple TFs [121,122]. Variation in the affinity for any given TFBS, the TF concentration, gene expression of the TF itself, and stoichiometry of the pool of TFs are important aspects of downstream transcriptional regulation for target genes [123].

For TFs and RNA polymerase/σ factor complexes to bind DNA, the DNA must be open and not occluded by other proteins. *Wolbachia* encode for three additional proteins which are predicted to bind DNA and putatively structure chromatin: a single-stranded DNA binding protein (known to stabilize single-stranded DNA in *E. coli* [124]), a few instances of an unidentified DNA binding protein, and HU (Heat Unstable), a histone-like protein found in bacteria that can also have effects on gene regulation [125] (Figure 2). *Wolbachia* also encode for a suite of TFs (Figure 2, Appendix A), several of which are present across all or nearly all *Wolbachia* strains surveyed, and others which are highly variable across the *Wolbachia* phylogeny. There are two putative TFs which are present in at least single copy across all strains: a YebC/PmpR-like TF and BolA. A YebC/PmpR homolog was previously identified as a Type III Secretion System (T3SS) regulator in the pathogen *Edwardsiella* (Enterobacteriaceae) [126] and as a pathogenicity regulator in *Pseudomonas aeruginosa* [127]. In *Escherichia coli*, BolA has roles in cell morphology, growth, motility, and stress responses, regulating a large set of genes [128,129,130,131].

The HU DNA binding protein and a MerR TF were present in all *Wolbachia* except for *w*Oo and *w*Ov, indicating a Supergroup C-specific loss of these two proteins. There were seven other classes of TFs that were highly variable in their presence across *Wolbachia* strains. Many strains have pseudogenized versions of these TFs, which might be indicative of recent selective pressures and genome reduction. The largest class of TFs were Helix-Turn-Helix type TFs. While XRE and MerR are subclasses of HTH proteins present in a subset of the *Wolbachia* strains, there were additional HTH-domain containing proteins present in the majority of *Wolbachia*. Relatively few of these Helix-Turn-Helix TFs were annotated as pseudogenized or partial. Additionally, there is a large amount of variability in the size of these predicted HTH proteins (92–328 amino acids), likely indicating a range of functions.

After transcription is initiated, proteins involved in elongation and termination can regulate gene expression through a variety of mechanisms that interact with both -cis and -trans factors. Elongation ensures that the RNA polymerase complex continues to transcribe the mRNA molecule, and termination stops transcription. Anti-termination factors result in the RNA polymerase complex ignoring a termination signal that is present in the middle of a transcript, thus preventing premature termination. In prokaryotes, Gre proteins aid in elongation [135]. All *Wolbachia* strains encoded for one copy of GreA, which is involved in resolving paused RNA polymerase during transcription [135,136,137] (Figure 2). *Wolbachia*’s GreA protein has been previously explored as an anti-filarial target [138], as clearing *Wolbachia* infections has been a successful strategy for treating filariasis due to the obligate nature of *Wolbachia*-nematode symbioses [139]. In addition to GreA, all *Wolbachia* encoded for one copy each of the proteins Rho, NusA, NusB, and NusG, except for *w*DacB which appears to encode for two copies of NusA (Figure 2, Appendix A). Rho is the canonical terminator protein in prokaryotes that initiates Rho-dependent transcriptional termination [140]. However, there are other -cis and -trans factors that can also result in termination, which are Rho-independent (for example, example hairpin structures or riboswitches) [141,142,143].

NusA is multifunctional: it can stimulate pausing of the RNA polymerase complex leading to termination (for example, at hairpins), or NusA can result in antitermination when in complex with NusB and NusG [135,144]. The bimodal functionality of NusA allows for the switching of transcription to either include or exclude 3’ sequence, which is often involved in regulating the expression of genes in an operon or of overlapping ORFs. For example, the two genes that induce and rescue CI in *Wolbachia* (*cifA* and *cifB* [145]) are, at least in the *w*Mel strain, separated by a Rho-independent transcription terminator that is predicted to form a GC-rich hairpin [105]. NusA might initiate termination at the hairpin (and thus only transcribe *cifA*), but when in complex with NusB and NusG, the hairpin might be ignored or resolved, leading to antitermination and thus transcription of a monocystronic *cifA/cifB* mRNA [105]. Indeed, the expression of *cifA* is always equal to or higher than the level of *cifB* expression [105]. The stoichiometry of *cifA* and *cifB* expression significantly varies due to host age and sex [105], and the bimodal functionality of NusA is one hypothesis for how *Wolbachia* might regulate these dynamics. The ratio and concentration of CifA and CifB could have downstream effects on the strength of CI induction in sperm and the ability to effectively rescue CI in eggs.

## 6. Secretion Systems

*Wolbachia* encode for several systems used to transport proteins. I find evidence for four putatively functional secretion systems across *Wolbachia*: Sec, Tat, T1SS, T4SS. Additionally many *Wolbachia* strains encoded for a protein annotated as (or homologous to) a T2SS protein (Figure 3, Appendix A). However, there is some evidence that this T2SS protein is related to T4 pilus proteins, so it may in fact be part of the T4SS, which is derived from bacterial conjugation machinery [146]. Because of this discrepancy, this protein was left out of Figure 3, but the annotations are available in Appendix A

The first two systems, Sec and Tat, move proteins from the cytoplasm to the periplasm (or into the inner membrane). The Sec system is composed of Sec proteins (SecABDEFGY), YajC, SRP, FtsY, and signal peptidases [88] (Figure 1). All *Wolbachia* strains had one copy of each protein, with just a few examples of pseudogenizations or split open reading frames (ORFs) (Figure 3, Appendix A). In *w*Npa and *w*Nfe, there were two partial copies of FtsY (the SRP-docking protein), but this may be an assembly error resulting in split ORFs. The other partially present Sec system proteins were SRP in *w*DacA, and signal peptidase I in *w*Wb. In the *w*Oo strain, SecY was pseudogenized. It is likely that the Sec system is functional across most *Wolbachia*, and additional work is needed to verify the loss of proteins in in the handful of strains with partial or pseudogeized Sec components. The second secretion system that translocates proteins into the periplasm is the Tat system. All *Wolbachia* encoded for two or three Tat proteins, always with at least one copy each of TatA and TatC, which are required for Tat function [90]. Strain *w*Npa coded for two copies of TatC, and there were two sets of monophyletic clades that encoded for two copies of the TatA protein (the group composed of *w*Pip, *w*Bol1B, and *w*Aus, as well as the C Supergroup strains *w*Oo and *w*Ov) (Figure 3, Appendix A).

The second group of secretion systems are those that translocate proteins across the outer membrane. All *Wolbachia* encoded for three proteins constituting a complete T1SS: a T1SS permease/ATPase that putatively spans the inner membrane, HlyD (the membrane fusing protein that spans the periplasm, and TolC (the component that spans the outer membrane) (Figure 1 and Figure 3, Appendix A) [81]. Very little is known about T1SS function or substrates in *Wolbachia*, but the T1SS has been implicated in virulence in other pathogens including *Legionella* [147,148], *Rickettsia* [149], *Orientia* [150], and *Anaplasma* [151]. In the Rickettsiales, several of the secreted T1SS effectors are ankyrin repeat containing proteins, predicted to be involved in protein-protein interactions [149,150,151].

*Wolbachia*’s T4SS is perhaps the most well understood of all the systems discussed here. Previous studies have described the evolution of T4SS proteins and genomic organization [101,106,152], identified putative T4SS substrates [38,153] (including the first described *Wolbachia* effector protein [39]), and analyzed T4SS expression patterns in the host [154]. Several microbiology resources are available to study *Wolbachia’s* T4SS including a candidate effector library [40] and a heterologous expression system in *E. coli* that uses a chimeric VirD4 (the T4SS coupling protein) to determine whether or not a substrate is secreted [155]. Many functional studies focus on the *w*Mel strain that infects the model insect *Drosophila melanogaster.* Here, I identify putatively functional T4SS across 29 *Wolbachia* strains and show that pseudogenization is relatively common. However, T4SS proteins in Rickettsiales are often encoded redundantly [101], and *Wolbachia* undergo high levels of genomic re-arrangements [156], which together might explain the dynamics of T4SS gene gain and loss across *Wolbachia*. For example, approximately a third of all the pseudogenized or partial T4SS proteins are *virB2* (which codes for the pilin subunit), which is highly duplicated and present in up to six putatively functional copies in some strains (Appendix A).

In *w*Bm, T4SS operon promotors are bound by two XRE-like TFs that share homology with EcxR, a *Ehrlichia chaffeensis* T4SS regulator [70,71]. XRE-like TFs were identified across most of the *Wolbachia* strains (Figure 2, Appendix A), so it is plausible they are a relatively conserved feature of T4SS regulation in *Wolbachia*. It remains to be determined whether additional TFs also contribute to T4SS regulation, how regulation varies across strains, and what the TFs signals or ligands are.

## 7. When do *Wolbachia* Respond to Their Environment?

While we have limited functional data on the systems and pathways described above, there are several studies that clearly indicate *Wolbachia* modifies gene or protein expression in response to changes in the host environment. Transcription and translation are metabolically expensive processes, so limiting un-necessary gene expression is important. Additionally, many proteins can be toxic or have off-target effects and would be harmful to *Wolbachia* and/or the host if expressed under the wrong conditions [157]. A strain of *Wolbachia* that is native to *Drosophila melanogaster*, *w*MelPop, regulates gene expression during doxycycline-induced stress in a transinfected mosquito cell line [158]. In the *Drosophila melanogaster* host, there are significant changes in *Wolbachia* gene expression patterns that correlate with fly age, life stage, and sex [159]. Interestingly, genes showing sex-biased expression patterns were also differentially expressed according to host age, perhaps indicating they are coregulated. The two genes which induce and rescue CI also show clear patterns of expression dependent on host age and sex [105]. The dynamics of this are likely critical for the proper expression of CI in the host and avoiding any toxic effects of these proteins [36]. Genes with the most significant changes in expression level are those previously predicted to be involved in stress-responses and host-interaction. These include molecular chaperones, components of secretion systems, and Ankyrin repeat domain containing proteins [159]. Additionally, genes with these dynamic transcriptional patterns were also more likely to be evolutionarily conserved across *Wolbachia*, perhaps highlighting their importance in host colonization and manipulation across the genus [159].

In nematode-infecting *Wolbachia*, which are more reduced in genome size, coding content, and the number of sensing systems and transcriptional regulators, we still see changes in *Wolbachia* physiology that correlate with some aspect of host biology. For example, the *w*Bm *Wolbachia* symbiont of the filarial nematode *Brugia malayi* expresses proteins in a host-stage and host-sex dependent manner [160]. Similar patterns in protein expression were identified in the *w*Ov symbiont of *Onchocerca volvulus* [161]. Additionally, *w*Ov also had different gene expression patterns in male worm somas, versus in female worm gonads, further indicating gene regulation in accordance with host context [162]. Lastly, the filarial nematode *Dirofilaria immitis* and its *Wolbachia* symbiont both display dynamic and coordinated changes in gene expression across the life of the worm [163]. Interestingly, most of the clade specific losses were in the Supergroup C strains *w*Oo and *w*Ov (Figure 2, Appendix A), which, as mentioned, still regulate transcription and protein expression in response to the host context.

There are numerous other scenarios in which we might imagine advantages for *Wolbachia* to regulate gene expression patterns in response to the host environment. Many of the studies find that host age and sex are commonly correlated with changes in *Wolbachia* gene expression. *Wolbachia* are obligately associated with the female germline, so it is quite likely that *Wolbachia* respond to host cues involved in or present during gonad maturation and oogenesis. Not only is this important for successful transmission to the next generation, but also the timing and dosage of expression for reproductive manipulation genes is critical for recapitulating phenotypes (such as CI) in a transgenic model [164,165]. Additionally, while *Wolbachia* are obligately associated with the female germline, they also infect a range of other tissue and cell types in the host in non-random patterns that depend on the host and *Wolbachia* strain combination [166]. In holometabolous insects (i.e., those with complete metamorphosis including a pupal stage), not only are organ structures and cell physiology different across development, but the host insect often feeds on very different diets as a larva and as an adult, resulting in drastically different metabolite availabilities across the host lifespan. Furthermore, many insects overwinter: a physiological state that *Wolbachia* too must be able to navigate. It would be advantageous for *Wolbachia* to respond to specific changes in physiology across all these conditions to limit fitness costs for the host (upon which *Wolbachia* relies for transmission) while simultaneously enhancing colonization and transmission for *Wolbachia.* Finally, while *Wolbachia* may respond to host-specific changes in their environment, there are many other external stressors that *Wolbachia* might respond to, including temperature and antibiotics.

In addition to physiological and physical changes associated with host age, tissue, and diet, *Wolbachia* are in many cases capable of horizontal transfer to new hosts, especially across insects [167]. These horizontal transfers present a drastic change in environment (including the potential for time outside of a host, as well as the new host itself), and likely come with many challenges for *Wolbachia*. New challenges might include differences in host immune responses with which *Wolbachia* is not co-adapted, the presence of different metabolites, and differences in host structure. For example, ovaries across insect evolution have major differences in yolk, the interfollicular tissue, and the organization of connections between nurse cells and oocytes [168]. Finally, there is also the potential for other intracellular symbionts or pathogens present in a host (including other *Wolbachia* strains) competing for similar niches as a newly arrived *Wolbachia.*

## 8. Future Directions

Across *Wolbachia* strains we see a relatively high level of conservation in systems that would aid in sensing the environment, regulating gene expression, and exporting proteins to the host. These include the PleC/PleD and CckA/CrtA TCS, at least two σ factors, a diversity of TFs, a set of termination, anti-termination, and elongation factors, and four secretion systems. The high level of evolutionary conservation is highly indicative of their importance across *Wolbachia* and strong selective pressures to maintain these systems. Indeed, even in strains with high levels of pseudogenization and gene loss, these systems have been maintained [46]. There are a few clear paths for future studies that would significantly advance our understanding of *Wolbachia* biology and host-microbe interactions.

Firstly, all the structures and functions discussed here are highly reliant on bioinformatic predictions. Pseudogenizations and losses need to be verified, especially for strains with draft genome assemblies that have not been closed into a circle. More in depth bioinformatic analyses will be important for detecting the presence of other proteins involved in gene regulation. *Wolbachia* genomes have many unannotated or broadly annotated proteins [46] which will require more in depth analysis (including domain predictions, homology searches, and Markov models), along with functional validation. Verifying the expression of these genes and proteins *in vivo*, using heterologous expression systems, and *in vitro* analyses to study the biochemistry of these proteins are essential for understanding the role of these proteins in *Wolbachia* biology. While there are no genetic tools currently available for transforming *Wolbachia,* complementation experiments can be performed in other systems (e.g., *Agrobacterium* [169]). For example, does *Wolbachia* CtrA rescue a CrtA knockout in *Agrobacterium,* indicating similar functionality? Bacterial two-hybrid systems are useful tools for determining if proteins directly interact with each other [170], which can help identify regulatory partners and other protein-protein interactions.

Many signaling pathways (e.g., PleC/PleD) make or respond to second messengers, such as c-di-GMP. These signaling molecules can be detected and quantified in *Wolbachia* exposed to different environments to determine if there is a second messenger response [171]. Again, complementation experiments combined with c-di-GMP detection would be useful for determining if proteins such as *Wolbachia’s* PleD can make and degrade c-di-GMP. Very few studies have studied *Wolbachia* physiology, and metabolomic approaches will likely be fruitful in understanding the interplay between *Wolbachia* and host. Additionally, dual RNA-seq of host and *Wolbachia* transcriptomes, either alone or coupled to metabolomics, is also likely to be a fruitful direction for understanding the symbiosis [172].

Regardless of the level or type of environmental response *Wolbachia* is capable of mounting, *Wolbachia*’s regulatory networks, and how they vary across strains, are almost completely unknown. Bioinformatic approaches can be used to predict TFBS [173], and antibodies can be raised against *Wolbachia*’s TFs to determine promotor sites and putative regulons [174]. Additional experiments will be necessary to identify which signals TFs are responding to, when alternative σ factors are employed, and how elongation and termination are regulated to fine tune expression. Which environmental signals funnel into which responses, and which intermediate proteins and signaling molecules are required?

Finally, there are several computational approaches that will be useful for understating the evolution of *Wolbachia* more broadly. Here, I identify the conservation of systems as indicated by presence or absence. However, more fine-scale analyses of protein evolution and selection will give clues as to which regions of the protein are particularly important and which points in the *Wolbachia* phylogeny have undergone rapid evolution or dynamic changes in gene gain or loss. For example, the T4SS pilus (encoded by *virB2*) is an external structure, and thus detectable by the host. Is VirB2 under strong selection pressure? Are there clade- or host-specific selection pressures on these *Wolbachia* proteins? *Wolbachia* are known to undergo high levels of rearrangements and horizontal transfers of genes [156]. Do the phylogenies of these systems all track the strain phylogeny? Or, are there genes that have been transferred between *Wolbachia* strains? The T4SS is particularly dynamic regarding gains and losses (Figure 3). *Wolbachia*’s phage WO has been proposed as one method by which horizontal transfer might be achieved [42,105]. Identifying biases in genomic context might give clues to if and how genes are horizontally transferred. For example, the prophage WO region in *w*Mel includes several transcription factors [42] that are recovered here in variable copy number across strains, including some Helix-Turn-Helix domain containing proteins, and the TF MarR (Figure 2).

Even though *Wolbachia* have relatively reduced genomes and minimal numbers of TCS and secretion systems compared to other intracellular bacteria, they can alter the physiology of diverse hosts in many ways. Expanding our knowledge of the regulatory networks *Wolbachia* employs to initiate infections and manipulate host biology will facilitate the use of *Wolbachia* in insect management programs and our understanding of the evolution of intracellular infections.

## Figures and Tables

**Figure 1 genes-11-00813-f001:**
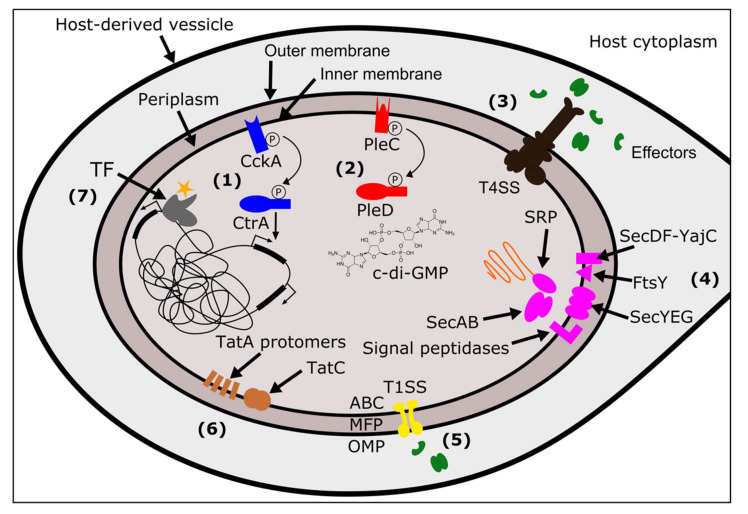
*Wolbachia’s* systems for sensing, responding to, and modifying the host. *Wolbachia* cells are surrounded by three membranes: two bacterial-derived membranes (inner and outer), and one host-derived membrane of golgi- or endoplasmic reticular-origin [31,85]. The host-derived membrane may totally enclose *Wolbachia*, or, it may connect to the rest of the host endomembrane system. Between *Wolbachia*’s inner and outer membranes is the periplasmic space. The TCS sensor histidine kinases detect changes in the periplasmic space and initiate phosphorelays. (**1**) The CckA/CtrA TCS, (blue), results in changes to gene expression of target genes. (**2**) The PleC/PleD TCS (red) regulates levels of c-di-GMP, a ubiquitous bacterial second messenger. (**3**) The T4SS (black) secretes effector proteins (green) across both *Wolbachia*-derived membranes. (**4**) The Sec system (pink) translocates proteins across or into the inner membrane. Unfolded proteins are brought to the SecYEG channel by either SecA/B or SRP/FtsY. SecDF-YajC facilitate movement through the channel and signal peptidases cleave the signal sequence within the periplasm. (**5**) The T1SS (yellow) is another secretion system that spans both inner and outer membranes for the secretion of proteins. An ABC transporter/ATPase (ABC) is embedded in the inner membrane, which connects to a membrane fusion protein (MFP) that crosses the periplasmic space. Finally, the outer membrane protein (OMP) spans the outer membrane to facilitate the last step of transfer out of the bacterial cell. (**6**) The Tat system (brown) translocates folded proteins to the periplasm using a complex of TatC and TatA proteins. (**7**) TFs (grey) modulate gene expression in response to a range of signals (orange star) including reactive oxygen species, heavy metals, the binding of other proteins, and various metabolites.

**Figure 2 genes-11-00813-f002:**
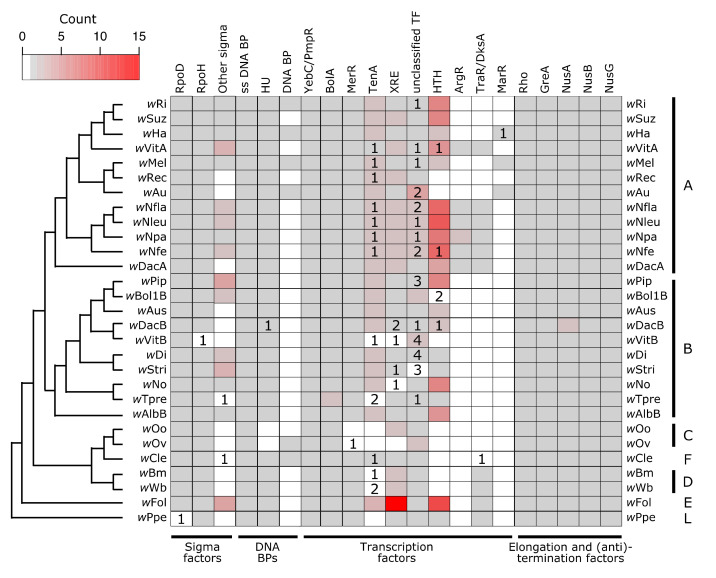
*Wolbachia* encode for a suite of proteins that are involved in transcriptional regulation. Cladogram was drawn based on previously published *Wolbachia* phylogenies to show relationships between strains [132,133,134]. Colors on the heatmap indicate how many putatively functional copies are present within a *Wolbachia* strain for a given gene, with white indicating no copies, and the grey to red color scale indicating between one and 15 copies. Numbers inside of the heatmap indicate how many additional copies of the gene are present but annotated as pseudogenized or partial. Supergroup membership (monophyletic clades of *Wolbachia* historically used for designation) of the strains is indicated with the capital letters on the right side of the heatmap. Abbreviations: ss = Single-Stranded; BP = Binding Protein; TF = Transcription Factor; HTH = Helix-Turn-Helix.

**Figure 3 genes-11-00813-f003:**
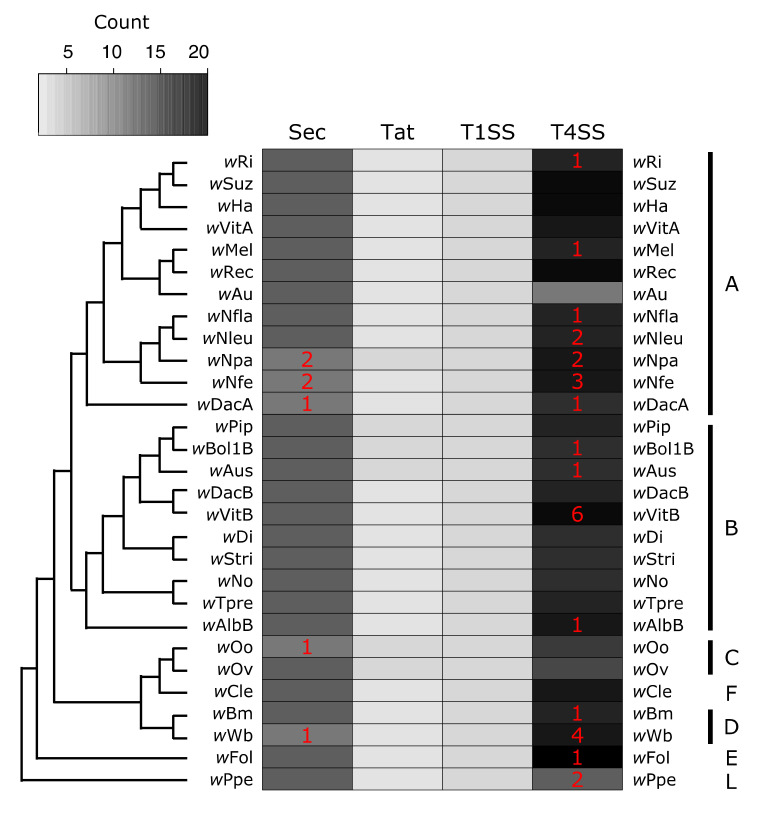
Presence of secretion systems across the *Wolbachia* phylogeny. Cladogram was drawn based off previously published *Wolbachia* phylogenies to show relationships between strains [132,133,134]. The grey to black scale indicates the number of putatively functional protein copies present within a *Wolbachia* strain for a given secretion system. Red numbers inside of the heatmap indicate how many additional gene copies are present but annotated as pseudogenized or partial (for example, *w*Ppe encodes for 12 proteins that make up a T4SS, plus an additional two pseudogenized ORFs also corresponding to T4SS proteins). Supergroup membership (monophyletic clades of *Wolbachia* historically used for designation) of the strains is indicated with the capital letters on the right side of the heatmap.

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
