# Peer review of "Sensing, Signaling, and Secretion: A Review and Analysis of Systems for Regulating Host Interaction in Wolbachia"

_genes, 2020, doi:10.3390/genes11070813_

Round 1
Reviewer 1 Report
Amelia R I Lindsey proposes an extensive review on the different systems that Wolbachia express within their host (sensing, signaling and secretion). She used genomic data from different databases, searched for the presence/absence of such systems and discussed their involvement in host-bacteria interactions. Overall, the review is very well written, and presents all the mechanisms in their context. This review will be very helpful for the scientific community working on host-Wolbachia interactions.
My main concern is about the treatment of the genomic data (very little information is given in the methods section): How to ensure that the analyses performed capture all the diversity of effectors present in each Wolbachia strain? For instance, which method is used (and which criteria are chosen?) to determine if a gene is considered as an ortholog? As a pseudogene? Were BLAST searches performed together with phylogenetic reconstructions? Indeed, if terms were only ‘grepped’ in the gene annotations from databases, many genes could miss in the screen.
Minor comments:
- l42: please explain other ways than reproductive manipulation through which Wolbachia can spread into population (ex: benefit (fecundity, protection…) higher than the cost associated with the presence of Wolbachia in host tissues)
- l53: please cite the fundamental work of Luis Teixeira’s team on the protective effect of Wolbachia on Drosophila (naturally infected by Wolbachia)
- l109: examples are given with extracellular bacteria. Would it be possible to give similar examples with intracellular bacteria?
- l132: please give details on the specificities of each secretion system (and on l169 on the T4SS)
- paragraph starting l142: Why so much emphasis is given to the Sec and Tat systems? Please equilibrate the amount of details given to present each system
- l230: Please make a new paragraph after ‘emerge’.
- l344-346: The introduction of this paragraph (transcriptional regulation) towards secretion systems is not the most obvious we could think about
- l372-374: please list some examples of virulence factors that are transported through this secretion system.
- l396: the figure is labelled figure 2 instead of figure 3
- l442-450: external stressors could also be included in this discussion
- l490-491: Dual RNAseq could also help to dig into the mechanisms involved in the interplay between Wolbachia and the host.
- It might be worthwhile to include in the discussion specific repeated genomic regions such as Octomom
Author Response
Please see my responses in bold.
Amelia R I Lindsey proposes an extensive review on the different systems that Wolbachia express within their host (sensing, signaling and secretion). She used genomic data from different databases, searched for the presence/absence of such systems and discussed their involvement in host-bacteria interactions. Overall, the review is very well written, and presents all the mechanisms in their context. This review will be very helpful for the scientific community working on host-Wolbachia interactions.
My main concern is about the treatment of the genomic data (very little information is given in the methods section): How to ensure that the analyses performed capture all the diversity of effectors present in each Wolbachia strain? For instance, which method is used (and which criteria are chosen?) to determine if a gene is considered as an ortholog? As a pseudogene? Were BLAST searches performed together with phylogenetic reconstructions? Indeed, if terms were only ‘grepped’ in the gene annotations from databases, many genes could miss in the screen.
Thanks for your feedback and close look at the article. Definitely searching for genes based on annotation alone would result in a lot of missed ORFs – I revises the text in the methods section to be clearer about how orthology was used to make sure we didn’t miss genes. Details to specific comments are below.
Minor comments:
- l42: please explain other ways than reproductive manipulation through which Wolbachia can spread into population (ex: benefit (fecundity, protection…) higher than the cost associated with the presence of Wolbachia in host tissues)
- I have added a few of the hypotheses for how Wolbachia might spread in a population.
- l53: please cite the fundamental work of Luis Teixeira’s team on the protective effect of Wolbachia on Drosophila (naturally infected by Wolbachia)
- This reference has been added.
- l109: examples are given with extracellular bacteria. Would it be possible to give similar examples with intracellular bacteria?
- I checked this section and it looks like references are made to both facultatively and obligately intracellular bacteria.
- l132: please give details on the specificities of each secretion system (and on l169 on the T4SS)
- I expanded these sections to give more details on the secretion systems.
- paragraph starting l142: Why so much emphasis is given to the Sec and Tat systems? Please equilibrate the amount of details given to present each system
- I opted to describe the Sec and Tat in more detail since T4SS (the more complicated of the systems that Wolbachia encode for) has been studied much more deeply and is discussed in other reviews that have a narrower focus than this one. As mentioned above, I added to the secretion system sections to make these more equal.
- l230: Please make a new paragraph after ‘emerge’.
- Done
- l344-346: The introduction of this paragraph (transcriptional regulation) towards secretion systems is not the most obvious we could think about
- I agree and have revised this section.
- l372-374: please list some examples of virulence factors that are transported through this secretion system.
- I have added this – examples are from the other species referenced as this has not been studied in Wolbachia.
- l396: the figure is labelled figure 2 instead of figure 3
- Fixed, thank you.
- l442-450: external stressors could also be included in this discussion
- This has been added.
- l490-491: Dual RNAseq could also help to dig into the mechanisms involved in the interplay between Wolbachia and the host.
- Absolutely, this has also been added.
- It might be worthwhile to include in the discussion specific repeated genomic regions such as Octomom
- I now include in another section of the manuscript the reference to the Octomom region and its importance as the first genotype-phenotype link.
Reviewer 2 Report
“Sensing, signaling, and secretion: A review and analysis of systems for regulating host interaction in Wolbachia” is an in-depth analysis of Wolbachia’s potential to interact with the environment. It provides a good overview of environment sensing, transcriptional regulation, and secretion systems found in Wolbachia genomes across phylogeny. I regard this manuscript as a good introduction to bacterial physiology for Wolbachia scientists, and a nice perspective for researchers working on other Rickettsiales. The most important insight, I believe, includes highlighting the fact that Wolbachia is not limited to T4SS for secretion, but also possesses T1SS for cytoplasm-to-environment transport.
Comments:
Line: 116-117: I am not sure about the division of responses into enzymatic or transcriptional. Negative regulators of signaling pathways which are proteins (but not enzymes) do not fit any of these categories. How about “protein level or transcriptional”? Or “gene expression or protein level”?
Line 126: Regulons are groups of genes regulated together. SS are “regulons of signaling pathways”? Could you please clarify? This is also a very long sentence.
Lines 142-160: What is the relevance of Sec and Tat systems in host-microbe interactions in bacteria without T2SS and T5SS?
Line 258: I am not sure what “DNA-binding sites, and signals” means here. Polymerase and sigma factors have their DNA-binding sites and are listed separately. Is it DNA sequences where proteins bind (cis and trans elements)? Are the signals small molecules (eg. lactose in lac operon activation) or other proteins (eg. repressor)?
Line 293: “Wolbachia encode three proteins which are predicted to bind DNA” – not true, as polymerases, sigma factors, TF, DNA repair proteins all bind DNA. More specificity is required, eg. calling them proteins structuring chromatin.
Line 396: There is an error in figures numbering: this is the second Fig. 2 in the paper.
Fig. 2: Most Wolbachia strains seem to have up to 15-20 complete gene sets coding for T4SS (and some additional single genes here and there) – is this correct? If my interpretation of this figure is correct, it is quite remarkable!
As the article is classified as a review, I believe it is important that it cites the literature correctly, including references to primary sources of data.
Here are some comments in this regard:
Line 46: I believe that a correct reference to Wolbachia prevalence in insects is both, the currently cited “Zug, R.; Hammerstein 2012” and also: Weinert LA, Araujo-Jnr E V., Ahmed MZ, Welch JJ (2015) The incidence of bacterial endosymbionts in terrestrial arthropods. Proc R Soc B Biol Sci 282(1807):20150249–20150249.
Line 53-54: Citations to seminal papers on Wolbachia genomics (eg. Wu et al. 2004), antiviral protection (Teixeira et al. 2008, Hedges et al. 2008, Moreira et al. 2009) or host factors (eg. Newton & Sheehan 2015) are missing here.
Line 58-61: The reference to the first link between genotype and phenotype in Wolbachia (Chrostek & Teixeira 2015) is missing.
Lines 404 – 420: Wolbachia respond to antibiotic-induced stress as well: Darby AC, et al. (2013) Integrated transcriptomic and proteomic analysis of the global response of Wolbachia to doxycycline-induced stress. ISME J:1–13.
Author Response
Please see my responses in bold.
“Sensing, signaling, and secretion: A review and analysis of systems for regulating host interaction in Wolbachia” is an in-depth analysis of Wolbachia’s potential to interact with the environment. It provides a good overview of environment sensing, transcriptional regulation, and secretion systems found in Wolbachia genomes across phylogeny. I regard this manuscript as a good introduction to bacterial physiology for Wolbachia scientists, and a nice perspective for researchers working on other Rickettsiales. The most important insight, I believe, includes highlighting the fact that Wolbachia is not limited to T4SS for secretion, but also possesses T1SS for cytoplasm-to-environment transport.
Thank you for the helpful feedback on the manuscript. Please see below for my responses to individual comments.
Comments:
Line: 116-117: I am not sure about the division of responses into enzymatic or transcriptional. Negative regulators of signaling pathways which are proteins (but not enzymes) do not fit any of these categories. How about “protein level or transcriptional”? Or “gene expression or protein level”?
Thank you for the suggestion – I have edited the text accordingly.
Line 126: Regulons are groups of genes regulated together. SS are “regulons of signaling pathways”? Could you please clarify? This is also a very long sentence.
I edited this sentence for clarity and length.
Lines 142-160: What is the relevance of Sec and Tat systems in host-microbe interactions in bacteria without T2SS and T5SS?
This is a great question – I added a few sentences to clarify this for the reader. Sec/Tat are important for embedding proteins into membranes – ex., two component system histidine kinases in the inner membrane. Or proteins can be localized to the periplasm and then other pathways are used to embed proteins in the outer membrane – which may interact directly with host factors.
Line 258: I am not sure what “DNA-binding sites, and signals” means here. Polymerase and sigma factors have their DNA-binding sites and are listed separately. Is it DNA sequences where proteins bind (cis and trans elements)? Are the signals small molecules (eg. lactose in lac operon activation) or other proteins (eg. repressor)?
Yes, this is correct. I updated the wording to be clearer.
Line 293: “Wolbachia encode three proteins which are predicted to bind DNA” – not true, as polymerases, sigma factors, TF, DNA repair proteins all bind DNA. More specificity is required, eg. calling them proteins structuring chromatin.
Yes, absolutely, this is misleading. I rephrased to say these are “additional proteins” that bind DNA – beyond the others already discussed in detail.
Line 396: There is an error in figures numbering: this is the second Fig. 2 in the paper.
Fixed, thank you.
Fig. 2: Most Wolbachia strains seem to have up to 15-20 complete gene sets coding for T4SS (and some additional single genes here and there) – is this correct? If my interpretation of this figure is correct, it is quite remarkable!
These Wolbachia strains encode for variable numbers of proteins that make up one T4SS – there is variation in the number and type of genes/proteins that are duplicated (ex., VirB2 is variable in copy number). I edited the wording so this would be clearer to the reader.
As the article is classified as a review, I believe it is important that it cites the literature correctly, including references to primary sources of data.
Here are some comments in this regard:
Line 46: I believe that a correct reference to Wolbachia prevalence in insects is both, the currently cited “Zug, R.; Hammerstein 2012” and also: Weinert LA, Araujo-Jnr E V., Ahmed MZ, Welch JJ (2015) The incidence of bacterial endosymbionts in terrestrial arthropods. Proc R Soc B Biol Sci 282(1807):20150249–20150249.
Line 53-54: Citations to seminal papers on Wolbachia genomics (eg. Wu et al. 2004), antiviral protection (Teixeira et al. 2008, Hedges et al. 2008, Moreira et al. 2009) or host factors (eg. Newton & Sheehan 2015) are missing here.
Line 58-61: The reference to the first link between genotype and phenotype in Wolbachia (Chrostek & Teixeira 2015) is missing.
Lines 404 – 420: Wolbachia respond to antibiotic-induced stress as well: Darby AC, et al. (2013) Integrated transcriptomic and proteomic analysis of the global response of Wolbachia to doxycycline-induced stress. ISME J:1–13.
Absolutely, thank you for checking these. All the aforementioned references and text to describe them were added.